# ARH1 in Health and Disease

**DOI:** 10.3390/cancers12020479

**Published:** 2020-02-19

**Authors:** Hiroko Ishiwata-Endo, Jiro Kato, Linda A. Stevens, Joel Moss

**Affiliations:** Pulmonary Branch, National Heart, Lung, and Blood Institute, National Institutes of Health, Bethesda, MD 20892-1590, USA; endohiro@mail.nih.gov (H.I.-E.); katoj@nhlbi.nih.gov (J.K.);

**Keywords:** arginine-specific mono-ADP-ribosylation, bacterial toxin, cholera toxin, ART1, ARH1, tumorigenesis, loss of heterozygosity, membrane repair, gender bias

## Abstract

Arginine-specific mono-adenosine diphosphate (ADP)-ribosylation is a nicotinamide adenine dinucleotide (NAD)^+^-dependent, reversible post-translational modification involving the transfer of an ADP-ribose from NAD^+^ by bacterial toxins and eukaryotic ADP-ribosyltransferases (ARTs) to arginine on an acceptor protein or peptide. ADP-ribosylarginine hydrolase 1 (ARH1) catalyzes the cleavage of the ADP-ribose-arginine bond, regenerating (arginine)protein. Arginine-specific mono-ADP-ribosylation catalyzed by bacterial toxins was first identified as a mechanism of disease pathogenesis. Cholera toxin ADP-ribosylates and activates the α subunit of Gαs, a guanine nucleotide-binding protein that stimulates adenylyl cyclase activity, increasing cyclic adenosine monophosphate (cAMP), and resulting in fluid and electrolyte loss. Arginine-specific mono-ADP-ribosylation in mammalian cells has potential roles in membrane repair, immunity, and cancer. In mammalian tissues, ARH1 is a cytosolic protein that is ubiquitously expressed. ARH1 deficiency increased tumorigenesis in a gender-specific manner. In the myocardium, in response to cellular injury, an arginine-specific mono-ADP-ribosylation cycle, involving ART1 and ARH1, regulated the level and cellular distribution of ADP-ribosylated tripartite motif-containing protein 72 (TRIM72). Confirmed substrates of ARH1 in vivo are Gαs and TRIM72, however, more than a thousand proteins, ADP-ribosylated on arginine, have been identified by proteomic analysis. This review summarizes the current understanding of the properties of ARH1, e.g., bacterial toxin action, myocardial membrane repair following injury, and tumorigenesis.

## 1. ARH Family

### 1.1. Properties of ARHs

ADP-ribosylarginine hydrolase 1 (ARH1) is a member of an ADP-ribosyl-acceptor hydrolase (ARH) family, which is composed of three 39-kDa proteins, ARH1-3, based on sequence and size similarity. These hydrolases differ in enzymatic activities and biological functions [1]. Human ARH1 is a 357 amino acid (aa) protein, which shares significant sequence and size conservation with ARH2 (354 aa protein, 47% identity, 68% similarity) and ARH3 (363 aa protein, 22% identity, 41% similarity). The mouse ARH1 amino acid sequence shows 83% identity and 91% similarity to human ARH1 [2]. ARH1 requires Mg^2+^ for catalytic activity [1,3,4]. The active sites of ARH1 (human, mouse, or rat) contain aspartates residues for coordination of Mg^2+^ binding [5,6]. Critical residues involved in ARH1 enzymatic activity and binding of ADP-ribose include the conserved vicinal aspartates 55 and 56 in humans [5,6] and 60 and 61 in rats [7]. Replacement of Asp 55 and/or Asp56 in humans and Asp60 and/or Asp61 in rats with alanine, glutamine, or asparagine significantly reduced hydrolase activity (ADP-ribosylarginine hydrolase, α-NADase) [5,6,7].

### 1.2. Function and Substrates of ARHs

Mono- and poly-ADP-ribosylation are reversible post-translational modifications of proteins, DNA and RNA [8,9]. The ARTs in mammals, i.e., ART1, ART2, ART5, are confirmed mono-ADP-ribosyltransferases that modify only arginine residues [10,11,12,13]. Target residues of the poly (ADP-ribose) polymerase (PARP) family include aspartate, glutamate, and serine [14,15,16]. Hydrolases, such as the ARHs and macrodomain proteins, also show amino acid-specific hydrolytic reactions. Mono-ADP-ribosylation on arginine, serine, and glutamate, respectively, was hydrolyzed by ARH1, ARH3, and macrodomain proteins, i.e., MacroD1, MacroD2, and terminal ADP-ribose protein glycohydrolase (TARG1)/C6orf130 [5,17]. Poly(ADP-ribose) glycohydrolase (PARG) cleaves poly(ADP-ribose) (PAR) chains of PARP-1 but did not cleave terminal ADP-ribose linked directly to amino acids of PARP-1. ARH3 releases ADP-ribose from serine on PARP-1 [18,19], but did not hydrolyze ADP-ribose-glutamate/aspartate linkages [16,20]. ADP-ribosylated DNA and RNA were hydrolyzed by ARH3, but not ARH1 [5,8,9]. Furthermore, ARH1 and ARH3 hydrolyzed the α-*O*-glycosidic bonds of the poly-ADP-ribose polymer attached to PARP1, although ARH3 has approximately 50 times more activity than ARH1 [1,21,22]. In addition, ARH1 and ARH3 hydrolyzed *O*-acetyl-ADP-ribose (*O*AADPr), the product of sirtuin deacetylases [23]. Recent reports showed that ARH2 is primarily expressed in the heart and appears to be involved in the regulation of heart chamber outgrowth [24]; although ARH2 has significant amino acid identity and homology to ARH1, it has not been shown to be an active hydrolase [2].

*Arh1*-deficient mouse embryonic fibroblasts (MEFs) and mouse tissues are unable to hydrolyze the ADP-ribose-arginine bond, supporting the view that ARH1 is the only mammalian arginine-specific hydrolase [25,26]. ARH1 has a role in controlling TRIM72 ADP-ribosylation, and thereby TRIM72 activity in membrane repair: Membrane injury may cause the mixing of intracellular with extracellular factors, e.g., small molecule such as NAD^+^, and proteins [27]. Another opportunity for ARH1 interaction with ART1 substrates might be through lipid raft-mediated endocytosis, which is enhanced by UV light and H_2_O_2_-induced stress [28]. Interestingly, glycosylphosphatidylinositol (GPI)-anchored ARTs and caveoline-3, a cofactor of TRIM72, exist within lipid rafts [10,29]. To address the question of how cytoplasmic protein ARH1 contributes to the reversion of arginine-ADP-ribosylation catalyzed by extracellular enzyme ARTs, localization of arginine ADP-ribosylated proteins needs to be determined.

Arginine-specific ADP-ribosylation has been identified in the endoplasmic reticulum (ER) protein binding immunoglobulin protein (BIP) also known as 78-kDa glucose-regulated protein (GRP78) [30], mitochondrial protein heat shock protein 75 (HSP75) also known as tumor necrosis factor receptor-associated protein 1TRAP1 [31], secreted protein tumor necrosis factor-α [32], and external plasma membrane protein integrin α7 [31], although, none of them has been identified as a substrate of ARH1. Under stress conditions, BIP/GRP78 and HSP75/TPAP1 are released to the cell surface and cytoplasm, respectively [33,34]. By this mechanism, cytoplasmic protein ARH1 might be able to interact with its target proteins. However, there is still the question as to how ecto-enzyme ARTs modify intracellular proteins. Further studies are needed to understand (1) whether extracellularly modified substrates can be internalized into cytoplasm and cleaved by ARH1, and (2) whether an intracellular transferase catalyzes arginine ADP-ribosylation.

Not all arginine ADP-ribosylated proteins have been reported to be hydrolyzed by ARH1. For example, human neutrophil peptide-1 (HNP-1) is ADP-ribosylated by ART1. HNP-1 is an antibacterial peptide contributing to host defense immunity, as well as being toxic for host epithelial cells. HNP-1 ADP-ribosylated on arginines 14 and 24 shows reduced antimicrobial and cytotoxic activities [35]. Non-enzymatic replacement of the ADP-ribosylated arginines of HNP-1 with ornithine resulted in a peptide with less cytotoxicity than unmodified HNP-1 but with retention of its antibacterial activity [36]. Thus, the ornithine-containing HNP-1, which is no longer a substrate for ARH1, may have therapeutic potential as an ARH1-resistant molecule.

Proteomic analysis has revealed that more than 1000 proteins are ADP-ribosylated on arginine residues by ARTs [31,37,38,39], the confirmed substrate proteins of ARH1 in vivo are only ADP-ribose-Gαs [26] and ADP-ribose-TRIM72 [27]. The amount and duration of cellular ADP-ribosylation may be controlled by substrate-specific transferases and hydrolases, which catalyze opposing arms of ADP-ribosylation cycles. Both ADP-ribosylation and de-ADP-ribosylation alter protein activities involved in cell signaling and viability [10,27,40].

## 2. Structure and Enzymatic Activity of ARH1

### 2.1. Structure and Stereospecific Activity of ARH1

ARH1 hydrolyzes the *N*-glycosidic linkage of ADP-ribosyl-arginine (e.g., arginine-Gαs, arginine-TRIM72) [10,26,27] and also cleaves the *O*-glycosidic linkage of poly-ADP-ribose, *O*-acetyl-ADP-ribose, and α-NAD^+^ [1,21,22,23]. The crystal structure of *h*ARH1 is available in a complex with ADP-ribose [5,6]. ARH1 requires Mg^2+^ for maximal catalytic activity [1,4,7]. The rat, mouse, and turkey ARH1 activities were stimulated by Mg^2+^ and dithiothreitol (DTT), whereas pig and calf ARH1 showed Mg^2+^, but not DTT dependence. Human ARH1 was DTT independent, as well [3,25]. The difference in DTT dependence appears to reside in the number of cysteine residues in the ARH1 proteins. ARH1 has a high structural similarity with ARH3 [6]. One major difference between ARH1 and ARH3 is the binding of the adenosine ribose moiety of ADP-ribose, resulting in a difference in ARH1 binding for ADP-ribose, which is 70× less than ARH3 [6]. ARH1 has four phosphorylation sites, tyrosine (Tyr)-4, Tyr-19, Tyr-20, and Tyr-205 [41]. Phosphorylation of ARH1 leads to conformational changes of the catalytic pocket, facilitating ADP-ribose-arginine binding to ARH1, which may affect ARH1 hydrolytic activity [41]. We speculate that the different affinities for ADP-ribose between ARH1 and ARH3 might be due to phosphorylation at the ADP-ribose binding pocket. ARH1 phosphorylation, at Tyr-4 and Tyr-19, was identified in a highly metastatic hepatocellular carcinoma (HCC) cell line, MHCC97H, but not in a nonmetastatic HCC cell line, Hep3B, implying that tyrosine phosphorylation of ARH1 was associated with HCC metastasis [42].

Mammalian ART1 catalyzes the transfer of ADP-ribose from β-NAD^+^ to proteins in a stereospecific manner, forming α-ADP-ribosylated-(arginine) proteins, which alters the protein’s function in cellular biological processes [43]. Arginine-specific ADP-ribosylation is a reversible modification, however, in vitro, α-ADP-ribosyl-arginine anomerizes to the β-form, which is not hydrolyzed by ARH1. Anomerization of α- to β-ADP-ribosyl-arginine interrupts a stereospecific ADP-ribosylation cycle in vitro [17,44,45,46]. Mono-ADP-ribosylated proteins are de-modified by the stereospecific α-ADP-ribose acceptor hydrolases, e.g., ADP-ribosyl-acceptor hydrolases (ARHs), MacroD1, MacroD2, Af1521, TARG1/C6orf130 [5,17]. ARH1 is the only ADP-ribosylated arginine-specific hydrolase identified in mammals that cleaves the *N*-glycosidic bond at the C1 linkage of α-anomeric ADP-ribose-arginine [4,17,25]. Currently, ARH1 has four stereospecific hydrolytic activities using as substrates α-NAD, α-*O*AADPr, poly(ADP-ribose), and α-ADP-ribose-(arginine) protein and generating free ADP-ribose as a product.

### 2.2. ARH1 Protein Expression and Cellular Distribution 

Among rat tissues, ADP-ribosylarginine-specific hydrolase activity was greatest in the brain, spleen, and testis [25]. ARH1 is a cytosolic protein, the product of a single gene that is ubiquitously expressed in mammalian tissues [25]. Similar to the finding with glycosylphosphatidylinositol (GPI)-linked ART1 protein expression [47], the amount of ARH1 protein increased during C2C12 myoblast differentiation into myotubes, indicating that it may play a role in myoblast differentiation [27]. ARH1 protein levels in lung adenocarcinoma and lymphoma in *Arh1^+^*^/*−*^ mice were lower than detectable levels by Western blotting. However, ARH1 existed in surrounding *Arh1^+^*^/*−*^ nontumorous lung tissue [40], suggesting that the loss of ARH1 activity enhanced tumor formation. These data are consistent with a role for inactivation or loss of the functioning *Arh1* gene or protein in the mouse tumorigenesis model. According to the human cancer database Oncomine (www.oncomine.org) [48], *ARH1* mRNA expression in human lung adenocarcinoma was significantly lower than in that of normal lung tissue [40,49], consistent with a tumor-suppressor function of ARH1.

### 2.3. Arginine-Specific Mono-ADP-Ribosylation Cycle

Arginine-specific ADP-ribosyltransferases and ADP-ribosylarginine hydrolase 1 (ARH1) are opposing arms of a mono-ADP-ribosylation cycle [50]. ADP-ribosylation of arginine was first discovered as a mechanism of action of bacterial toxins, e.g., *vibrio cholerae* cholera toxin, *Escherichia coli* (*E. coli*) heat-labile enterotoxin, *pseudomonas aeruginosa* exoenzyme S (ExoS), which catalyze the NAD^+^-dependent disruption of the signal transduction pathway by ADP-ribosylation of critical proteins, e.g., G protein alpha subunit that is stimulating for adenylyl cyclase (Gαs), rat sarcoma viral oncogene homolog (Ras), and Ras-related protein in brain (Rab) [51,52,53,54]. Cholera toxin produces arginine ADP-ribosylated Gαs, resulting in intoxication, which is terminated by ARH1 cleaving ADP-ribose from ADP-ribosylated Gαs. Thus, arginine-specific mono-ADP-ribosylation is a reversible reaction.

Other possible substrates of ARH1 might be Ras and Rab. It has been reported that ExoS catalyzes arginine ADP-ribosylation of Ras and Rab inhibiting nerve growth factor-stimulated neurite formation of PC-12 cells and disrupting normal vesicle trafficking, respectively, however, it has not known whether ARH1 cleaves ADP-ribose from ADP-ribosylated Ras and Rab [53,54].

In mammalian cells, endogenous ADP-ribosyltransferases (ARTs), extracellular GPI-anchored ART1 and ART2, and secreted ART5, catalyze arginine-specific ADP-ribosylation similar to those of the bacterial toxins, e.g., cholera toxin, *E. coli* toxin [10,11,12,13]. Ecto-ART proteins show tissue-specific expression such as heart and skeletal muscle for ART1, lymphocytes for mouse ART2, and testis for ART5. In humans and chimpanzees, however, ART2 is a pseudogene [10]. Furthermore, ART5 is primarily an NAD^+^ glycohydrolase; NADase activity of ART5 is 10x higher than its ADP-ribosyltransferase activity [10,55]. Therefore, GPI-linked ART1 seems to be a primary contributor to arginine-specific ADP-ribosylation in humans [35,56]. Under normal conditions, there is no difference in NAD^+^ levels of lung, heart, and brain between wild-type and *Arh1*-deficient mice, suggesting that ARH1 does not consume NAD^+^ [49]. However, ARH1 decreased the levels of ADP-ribose-(arginine) content [27]. Arginine-specific ADP-ribosyltransferase activity in mouse heart did not differ between wild-type and *Arh1*-deficient mice, suggesting that the accumulation of ADP-ribosylarginine content is due to ARH1 deficiency in *Arh1^−/−^* mice [27]. These data imply that cardiomyocytes are undergoing an arginine-specific ADP-ribosylation cycle in vivo. It is not surprising that this hypothesis raises questions regarding how an extracellular protein, GPI-linked ART1, catalyzes ADP-ribosylation in the extracellular space where the NAD^+^ concentration is 0.1 µM [11,13,57] and how cytoplasmic protein ARH1 hydrolyzes ADP-ribosyated proteins synthesized by an extracellular enzyme ART1. There is evidence that cellular NAD^+^ may be released into the extracellular matrix during inflammation and under pathological circumstances where cells may be killed by ischemic stress, thus providing substrate for the ADP-ribosyltransferases [58,59,60]. Additional evidence suggests that serum TRIM72 is released from injured or dead cells and is detectable following muscle injury induced by cardiac ischemia-reperfusion and treadmill exercise [61,62]. Furthermore, ART1, ARH1, caveolin-3, and cytoplasmic membrane repair protein TRIM72 were detected in macromolecule complexes [27]. Altogether, cytoplasmic ARH1 may leak with TRIM72 and NAD^+^ into the extracellular space where ART1 resides. In the last step of the cycle, the release of ADP-ribose from ADP-ribosylated TRIM72 by ARH1 promotes oligomerization of TRIM72 and recruitment of TRIM72 to the site of injury [27]. Thus, ART1-TRIM72-ARH1 appears to constitute an ADP-ribosylation cycle.

## 3. Functions of ARH1 in Disease

### 3.1. Defense Mechanism against the Action of Cholera Toxin

According to the World Health Organization (WHO), each year, there are still about 1.3 million to 4 million cases of cholera and 21,000 to 143,000 deaths worldwide [63]. Cholera toxin produced by *Vibrio cholerae* consists of a catalytic A-subunit, which dissociates from its B-subunits in the ER and catalyzes ADP-ribosylation of the α subunit of the intestinal Gs protein (Gαs) [64,65]. ADP-ribosylated Gαs at arginine 187 is incapable of hydrolyzing GTP and remains in an active state, resulting in stimulation of adenylyl cyclase (AC) and increased cyclic AMP (cAMP) formation. In this model, increased cAMP activates the cystic fibrosis transmembrane conductance regulator (CFTR) chloride channel, which leads to a loss of Cl^−^, Na^+^, and water in the intestinal lumen, causing the devastating diarrhea characteristic of cholera [26,64,66,67].

Cholera toxin ADP-ribosylates arginine moieties in a number of proteins, e.g., unidentified 18-, 98-, and 200-kDa proteins, however, Gαs including Gαs-S and Gαs-L appears to be the predominant protein that is ADP-ribosylated on arginine by cholera toxin [68]. The amount of ADP-ribosyl Gαs in the presence of cholera toxin was greater in *Arh1*-deficient mice, which suggests that, in wild-type mice, ARH1 cleaves ADP-ribose from Gαs, thereby generating unmodified Gαs and reducing fluid accumulation caused by cholera toxin [26]. Indeed, *Arh1*-deficient mice were more sensitive to cholera toxin-stimulated fluid accumulation in intestinal loops than wild-type mice [69]. In addition, the ARH1-based host defense mechanism occurs in a gender-specific manner. Female *Arh1*-deficient mice were more sensitive to the cholera toxin than were male mice [69]. Male and female wild-type mice, however, did not show a difference in cholera toxin sensitivity. The knockout mice but not the wild-type mice data supported the finding that women had a higher prevalence of cholera than men [70]. In addition to differences in ARH1 in humans and mice, these gender effects in humans may result from other factors such as societal norms (e.g., domestic responsibility for caring of the sick, time spent at home, and accessibility to health care), rather than biological differences in reaction to cholera and/or cholera toxin.

### 3.2. Tumor-Suppressor Function of ARH1

#### 3.2.1. Increased Tumor Formation in *Arh1*-Deficient and *Arh1*-Heterozygous Mice

Increased tumorigenesis was seen in *Arh1*-deficient and *Arh1*-heterozygous mice [2,40]. During a 24-months observation period, 20.5% (32 out of 156 mice) of *Arh1*-deficient and 11% (19 out of 169 mice) of *Arh1*-heterozygous mice showed increased frequency and extent of tumors in multiple organs, e.g., adenocarcinoma in lung, uterus, and mammary gland; hepatocellular carcinoma; hepatic and gastrointestinal lymphoma; hemangiosarcoma [40]. Tumors between *Arh1*^−/−^ and *Arh1*^+/−^ mice differed in the age of appearance with *Arh1*^−/−^ and *Arh1*^+/−^ mice, showing tumors at 3 months and 6 months, respectively [40]. Consistent with increased in vivo tumorigenesis, mouse embryonic fibroblasts (MEFs) generated from *Arh1* knockout and heterozygous mice showed increased cell proliferation and tumor formation in nude mice compared to wild-type MEFs [40]. Furthermore, *Arh1*-knockout MEFs transformed with an inactive double-mutant (D60, 61A) *Arh1* gene [7,40] did not rescue the *Arh1* knockout MEFs and showed increased cell proliferation as well as tumor formation in nude mice [2]. In agreement, overexpression of active ARH1 protein in *Arh1*-deficient MEFs partially reversed the tendency to develop tumors [2]. Other *Arh1*^+/−^ MEFs that developed tumors in nude mice showed loss of heterozygosity of the remaining *Arh1* gene. Tumorigenic MEFs with *Arh1* gene heterozygosity showed a mutation in the remaining allele and expressed a low level of ARH1 activity [2]. These data are consistent with a tumor-suppressor function of ARH1. 

The transmembrane ecto-enzyme CD38 functions as a NAD glycohydrolase and an ADP-ribosyl cyclase and is an NAD^+^-dependent oncogene [71]. Consistent with this hypothesis, CD38 was overexpressed in 41% (11 out of 27 human lung tumor samples) of tumor cells [49]. The anti-CD38 monoclonal antibody, daratumumab (DARA), is an approved treatment for patients with multiple myeloma [72]. CD38 activities were inhibited by ADP-ribosylation on arginine [71]; it is not known whether ADP-ribosylated CD38 is a substrate of ARH1. Deletion of the *Cd38* gene reduced tumor formation in both *Arh1*-deficient and wild-type mice [49], with significant reductions in the incidence of lymphomas, adenocarcinoma, and hemangio/histolytic sarcomas [49]. Knockout of CD38 in A549 human adenocarcinoma cells inhibited anchorage-independent cell growth, cell invasion, and xenograft growth in nude mice [49]. In contrast, *Arh1*-deficiency in MEFs affected cell cycle progression, resulting in increased cell proliferation [40]. These data suggest that ARH1 affects the cell cycle, preventing tumor formation, rather than control cell migration, as is the case with CD38-mediated metastasis [22,49]. In addition, estrogen promoted the survival rate of *Arh1*-deficient MEFs in the murine circulation and increased tumor metastasis to the lung [73]. As described above, increased tumor formation was dependent on the loss of ARH1 activity. Thus, ARH1 plays a role in cell proliferation in response to modifiers of tumorigenesis, e.g., CD38, estrogen.

#### 3.2.2. ARH1 Heterozygosity and Tumorigenesis

As noted earlier, a 24-month observation of ARH1 littermates from birth revealed that *Arh1*-deficient mice showed a 1.8× higher incidence of tumor formation than *Arh1*-heterozygous mice. However, between the ages of 24 and 33 months, the frequency of tumors seen in *Arh1*-deficient and heterozygous mice was similar, 31% and 28%, respectively. This age-dependent increased occurrence of malignancy in *Arh1*-heterozygous mice resulted in a loss of heterozygosity (LOH) and an absence or mutation of the *Arh1* gene. Mutation of the good allele in the ARH1 heterozygous mice resulted in an ARH1 protein whose activity was between 4% to 55% of the wild-type ARH1 [2,40]. *Arh1* gene mutation in MEFs and *Arh1*^+/−^ heterozygous mice tended to be in exons 2 and 3 that is comparable to the human ARH1 catalytic site in exons 3 and 4 [2,40]. In the human cancer database, LOH of the *ARH1* gene was identified in the lung (15%) and kidney (18%) [2]. According to the human somatic tumor mutation database, human *ARH1* gene mutations observed in cancer were also located in the human ARH1 catalytic region that corresponds to the mutation sites in mouse tumors [2]. Based on these findings, ARH1 in the murine model appears to be applicable to human cancer studies. Together, these data support the hypothesis that ARH1 is a tumor-suppressor gene that participates in the pathogenesis of both human and mouse cancers.

### 3.3. Membrane Repair Function of ARH1 

*Arh1*-deficient 8-month-old mice developed cardiomyopathy with myocardial fibrosis. Cardiac fibrosis occurred in a gender-specific manner with fibrosis in *Arh1*-knockout male mice being 10× greater than in female mice [27]. Cardiac fibrosis is characterized by increased collagen type I deposition due to aging or as a result of injury, e.g., myocardial infarction, hypertensive heart disease, idiopathic dilated cardiomyopathy, and diabetic hypertrophic cardiomyopathy [74]. In contrast, regardless of sex, during dobutamine-induced stress, *Arh1*-knockout mice showed significantly lower ejection fraction and fractional shortening than wild-type mice, consistent with systolic dysfunction [27]. The membrane repair protein TRIM72 was identified as a substrate for ARH1 and ART1 [27]. TRIM72 had been described as an essential molecule of the membrane repair process, recruiting intracellular vesicles to sites of membrane disruption [75,76]. Cytoplasmic protein TRIM72 leaks from injured cardiac tissue into serum [61], thus serum TRIM72 may be a potential biomarker of acute cardiac injury. TRIM72 was ADP-ribosylated on arginines 207 and 260 [37,39]. The endogenous ADP-ribosylated TRIM72 level was elevated in *Arh1*-deficient mice following cardiac ischemia-reperfusion injury. 

In rat cardiac myocytes, greater than 80% of cellular NAD^+^ is in mitochondria [77]. Indeed, NAD^+^ release from mitochondria in cytosol protects myocytes from post-ischemic reperfusion injury [77]. The importance of ARH1, ART1, and TRIM72 ADP-ribosylation cycle in plasma membrane repair and wound healing was demonstrated using a laser injury model and scratch wound-healing assay in C2C12 myotubes after stable transformation with TRIM72, ARH1, ART1, and ARH1 plus ART1 shRNA, and transient transformation with wild-type TRIM72-GFP and double mutant TRIM72 (R207K, R260K)-GFP that is not ADP-ribosylated [27]. In addition, heterogeneous complexes containing TRIM72 with components of a reversible ADP-ribosylation cycle included ART1, ARH1, caveolin-3 [27,75,78,79]. Notably, the mono-ADP-ribosyltransferase inhibitors vitamin K_1_ and novobiocin, as well as the loss of ARH1 activity, inhibited the oligomerization of TRIM72, the essential mechanism by which TRIM72 is recruited to the site of membrane injury [27]. Taken together, the arginine mono-ADP-ribosylation cycle controlled by ART1 and ARH1 is fundamental to the oligomerization of TRIM72 during the membrane repair process in cardiomyocytes (Figure 1).

## 4. Proteomics

Reanalysis of phosphoproteomic data identified arginine mono-ADP-ribosylation sites on 79 proteins [37]. Using a peptide-based enrichment strategy, 830 proteins containing ADP-ribosylated arginine were identified in mouse liver after H_2_O_2_-induced oxidative stress [31]. Arginine was the major (86%) ADP-ribosylated amino acid in mouse liver lysed in RIPA buffer with 40 µM PJ-34 and 1 µM adenosine diphosphate (hydroxymethyl) pyrrolidinediol (ADP-HPD), a specific PARP inhibitor (IC_50_ 120 nM) [6,31]. Following overexpression of murine arginine-specific ADP-ribosyltransferase 2 (ART2) in microglia, 33 arginine ADP-ribosylated proteins were identified [38]. Most recently, 354 proteins ADP-ribosylated on arginine were identified from wild-type mouse heart, mouse skeletal muscle, and C2C12 myotubes (Figure 2) [39]. Six ADP-ribosylated proteins overlapping with mouse heart, mouse skeletal muscle, and C2C12 myotubes were identified (Figure 2), e.g., Golgi apparatus protein 1 (Glg1), basement membrane-specific heparan sulfate proteoglycan core protein (Hspg2), integrin α-7, nidogen-1, protein disulfide-isomerase A3, and TRIM72 (Table 1). HNP-1, integrin α-7, and TRIM72 represent the only previously reported ART1 target proteins [5,27,36,80,81]. Among these 354 proteins, only 6 proteins were identified in *Art1*-deficient mouse skeletal muscle (Table 2) [39]. These data support the hypothesis that ART1 is the main contributor to skeletal muscle and heart arginine ADP-ribosylation.

Together, recent advances in proteomic analysis uncovered more than 1000 proteins ADP-ribosylated on arginine residues [31,32,37,38,39]. ADP-ribosylome data, analyzed by the gene ontology, suggested that ADP-ribosylated proteins of the wild-type mouse heart, but not *Art1*-deficient mouse heart (i.e., ART1-independent ADP-ribosylated proteins), are involved in the regulation of muscle contraction and apoptotic processes [39]. Further analysis by STRING-based protein-protein interaction analysis identified arginine ADP-ribosylated protein interaction networks that are involved in stress response, wounding response, and regulation of the heart rate [39]. Thus, arginine-specific ADP-ribosylation cycle controlled by ART1 and ARH1 is important in muscle physiology and pathophysiology.

## 5. Conclusions

Increasing evidence from knockout mouse models, where the ADP-ribosylation cycle is disrupted, shows the importance of arginine-specific ADP-ribosyltion cycles in disease, e.g., cancer [2,40], bacterial toxin-mediated infection [26,69], cardiomyopathy with myocardial fibrosis [27], and muscle weakness [39]. Data are consistent with ART1 and ARH1 serving as opposing arms of an arginine-specific ADP-ribosylation cycle. The COSMIC database analysis of human somatic mutations in cancer revealed 32 *ARH1* mutations in human lung, breast, and colon cancers, overlapping with the mutations found in *Arh1*-heterozygous mice after 6 months of age [2], demonstrating that ARH1 is an age-related cancer risk factor. In addition, *Arh1* deficiency resulted in gender-biased phenotypes. *Arh1*-deficient mice showed a female-biased increase in tumorigenicity and susceptibility to cholera [26,49,82]. In contrast, cardiomyopathy with myocardial fibrosis was seen in male more than female *Arh1*-deficient mice [27]. Recently, advances in mass spectrometry-based proteomics identified numerous arginine ADP-ribosylated proteins as well as the location of their modification sites in vitro. The role of the modification on function has been demonstrated for a limited number of proteins such as Gαs, TRIM72, and HNP-1, probably due to difficulties in reproducing mono-ADP-ribosylation, in vivo. Finding substrates of ARH1 can lead to studies of physiologically and pathologically relevant conditions for deciphering ARH1 functions in health and disease.

## Figures and Tables

**Figure 1 cancers-12-00479-f001:**
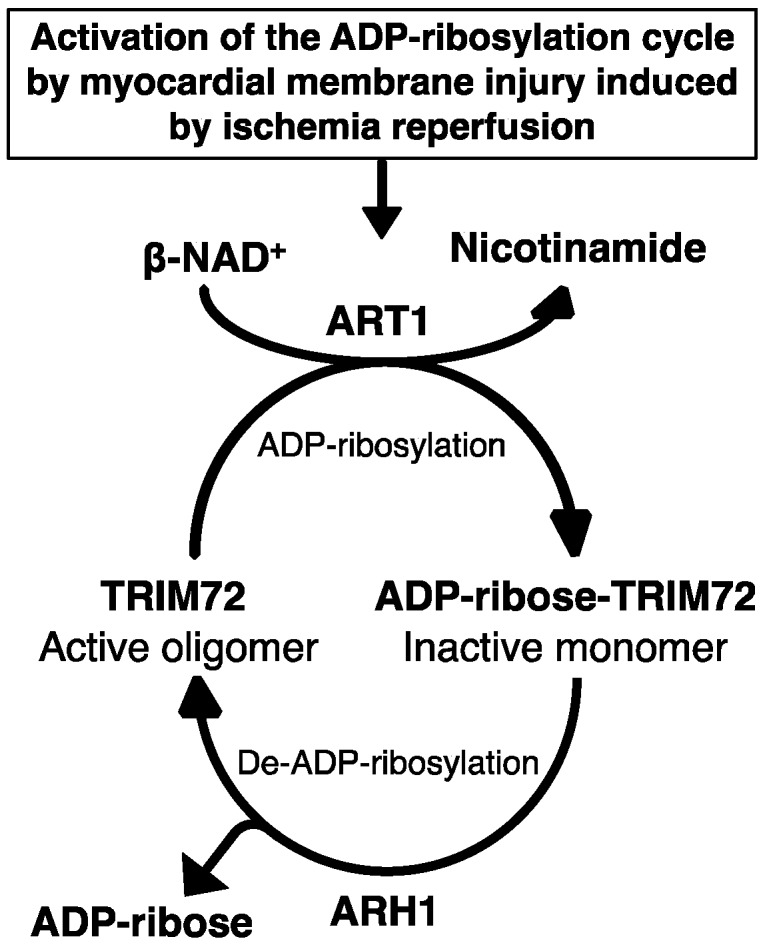
Tripartite motif-containing protein 72 (TRIM72) adenosine diphosphate (ADP)-ribosylation cycle in membrane repair. Ischemia-reperfusion-induced membrane disruption increased the ADP-ribosylation of TRIM72 by ADP-ribosyltransferase (ART) 1 at the sites of membrane damage, facilitating binding of TRIM72 and caveolin-3 to the membrane. Oligomerization of TRIM72 is essential for acute membrane repair and involves the recruitment of TRIM72 and intracellular vesicles at the injury sites [78]. ADP-ribosylarginine hydrolase (ARH) 1 catalyzes de-ADP-ribosylation of modified TRIM72, cleaving the ADP-ribose from ADP-ribosylated (arginine)TRIM72, promoting oligomerization of TRIM72 at the sites of injury.

**Figure 2 cancers-12-00479-f002:**
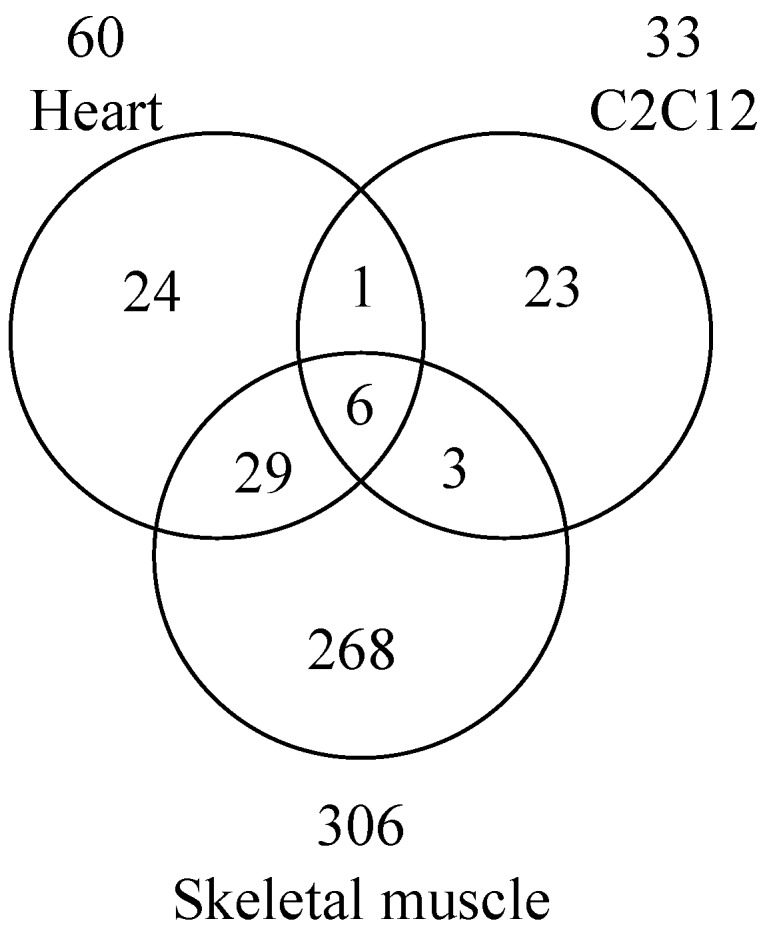
Venn diagram of proteins ADP-ribosylated on arginine in wild-type mouse heart, skeletal muscle, and C2C12 myotubes. Data are modified from Leutert et al. [39].

**Table 1 cancers-12-00479-t001:** Proteins adenosine diphosphate (ADP)-ribosylated on arginine residues in C2C12 myotubes, mouse skeletal muscle, and mouse heart. Data derived from Leutert et al. [39].

Gene Name	Sample	ADP-Ribosylation Sites	Protein Accesion	Protein Description
Glg1	C2C12 myotubes	R94	tr|F8WHM5|F8WHM5_MOUSE	Golgi apparatus protein 1 (Fragment)
	Skeletal muscle	R94, R313, R909, R1071	tr|F8WHM5|F8WHM5_MOUSE	Golgi apparatus protein 1 (Fragment)
	Heart	R94	tr|F8WHM5|F8WHM5_MOUSE	Golgi apparatus protein 1 (Fragment)
Hspg2	C2C12 myotubes	R588	tr|B1B0C7|B1B0C7_MOUSE	Basement membrane-specific heparan sulfate proteoglycan core protein
	Skeletal muscle	R588, R1956, R2957, R4018, R4148	tr|B1B0C7|B1B0C7_MOUSE	Basement membrane-specific heparan sulfate proteoglycan core protein
	Heart	R588	tr|B1B0C7|B1B0C7_MOUSE	Basement membrane-specific heparan sulfate proteoglycan core protein
Itga7	C2C12 myotubes	R149, R898	sp|Q61738-2|ITA7_MOUSE	Isoform Alpha-7X1A of Integrin alpha-7
	Skeletal muscle	R548	tr|G3X9Q1|G3X9Q1_MOUSE	Integrin alpha 7
	Heart	R608, R896	sp|Q61738-2|ITA7_MOUSE	Isoform Alpha-7X1A of Integrin alpha-7
Nid1	C2C12 myotubes	R318	sp|P10493|NID1_MOUSE	Nidogen-1
	Skeletal muscle	R318, R349, R799	sp|P10493|NID1_MOUSE	Nidogen-1
	Heart	R318	sp|P10493|NID1_MOUSE	Nidogen-1
Pdia3	C2C12 myotubes	R39, R62	tr|F6Q404|F6Q404_MOUSE	Protein disulfide-isomerase A3 (Fragment)
	Skeletal muscle	R39	tr|F6Q404|F6Q404_MOUSE	Protein disulfide-isomerase A3 (Fragment)
	Heart	R62	sp|P27773|PDIA3_MOUSE	Protein disulfide-isomerase A3
Trim72	C2C12 myotubes	R118	sp|Q1XH17|TRI72_MOUSE	Tripartite motif-containing protein 72
	Skeletal muscle	R115, R118, R207, R371	sp|Q1XH17|TRI72_MOUSE	Tripartite motif-containing protein 72
	Heart	R118, R260, R207	sp|Q1XH17|TRI72_MOUSE	Tripartite motif-containing protein 72

**Table 2 cancers-12-00479-t002:** Characterization of ADP-ribosylation by ADP-ribosyltransferase (ART)1 of mouse heart and skeletal muscle proteins. *Art1*-deficient mouse heart and skeletal muscle contained very few proteins ADP-ribosylated on arginine residues compared to the wild-type, which suggested that ART1 is a major contributor to ADP-ribosylation in skeletal muscle and heart. Data derived from Figure 3B from Leutert et al. [39].

Number of ADP-Ribosylated Protein		ADP-Ribosylated Amino Acid					
		Arg	Glu	Lys	Ser	Asp	Met
C2C12 cells	Wild-type	33	0	1	1	1	0
Skeletal muscle	Wild-type	303	0	2	1	1	0
	Art1-KO	6	2	2	4	1	0
Heart	Wild-type	60	5	5	7	2	1
	Art1-KO	0	4	4	7	2	0

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
