# Peer review of "ARH1 in Health and Disease"

_cancers, 2020, doi:10.3390/cancers12020479_

Round 1

Reviewer 1 Report

The review summarizes the role of ARH1 in the control of arginine ADP-ribosylation and its association with cell physiology and disease.

In general, many of the statements are not very clear and should be more carefully formulated (some examples are given below). Moreover, the discussion could be more to the point and certainly more critical. As it stands it reads like a list of findings that have been published without any conclusion. Of course this is useful but I expect more from the Moss lab as the experts on ARH1. One of the open questions, as far as I understand the review and also from reading a few papers on arginine-ADPr is that the enzymes performing the ADP-ribosylation reaction are all extracellular localized. Why are so many R-ADPr substrates described from mass spectrometry analysis but no one talks about the enzymes? One of the papers of the authors, ref 20 in the review, suggests that TRIM72 becomes a substrate of ART1, an extracellular enzyme, once the cell membrane is damaged. Is this part of the explanation that R-ADPr substrates are the result of damaged cells that then expose multiple substrates to extracellular enzymes? How clear is it that for the cellular and organismal phenotype, as in ref 20, the catalytic activity is relevant? While I can understand that ER resident proteins could be substrate, as ART1 passes through the ER, more difficult to understand are e.g. mitochondrial substrates (as mentioned in the listed mass spectrometry papers). Also in light, as pointed out in the review, that ARH1 is not found in mitochondria. All this is rather confusing to the non-specialist and I expect from the experts in the field that they provide a critical discussion of all these issues.

In the same direction, most paragraphs summarize large sets of data but no conclusions are drawn. This is necessary to make this review readable for non-experts. Of course, there may not be clear conclusions that can be drawn. This would also be important to mention.  

Page 3: stereospecificity is discussed. While there is evidence that alpha-NAD but not beta-NAD are hydrolyzed by ARH1 and other hydrolases. It is not becoming clear whether this has any relevance for substrate de-ADP-ribosylation.

Page 4: It is indicated that female Arh1 KO animals are more sensitive to cholera toxin than wt animals. This is not unexpected. But it is unclear why a point is made for female animals. Is there a difference to male animals? On the same page below a connection to estrogens is mentioned e.g. in relation to tumor formation in Arh1knockout animals. Unfortunately, no further information is given. What is the connection of estrogen to Arh1 or Arh1 substrates? Can any mechanistic interaction / model be suggested? Please discuss.

Further on this page is the statement “over 20% of Arh1-deficient and 11% of Arh1-heterozygous mice showed increased frequency and extent of tumors in multiple organs”. This is unclear. Should it mean that 20% / 11% of animals develop tumors compared to control?

Also, the statement “Furthermore, Arh1-knockout MEFs transformed with an inactive double-mutant (D60, 61A) Arh1 gene, showed increased cell proliferation and tumor formation as seen in the parent ARH1 knockout mice [2].” is not clear. Does it mean that the mutant has no additional effect, i.e. does not rescue? Or does it mean that it enhances the knockout phenotype? This should be clarified.

At the bottom of page 4 a functional interaction of CD38 and Arh1 is indicated. Again, some discussion of a potential mechanism for this would be useful for the reader.

Reviewer 2 Report

In this review Ishiwata-Endo and colleagues summarize data concerning ARH1 molecular functions.

In general, this review is a summary of the available data but is mainly lacking the critical interpretation of the findings. Further, it is stated that the major Arginine-ADP-ribosyltransferase is ARTC1, an ecto enzyme and as such catalyzing ADP-ribosylation extracellularly. In contrast ARH1 is exclusively found intracellularly. How can ARH1 contribute to the reversion of ARTC1 catalyzed Arginine-ADP-ribosylation? Are there any speculations about intracellular enzymes catalyzing the modification of Arginine by ADP-ribosylation? Are extracellularly modified substrates somehow internalized to get into contact with ARH1? To me these are obvious questions that are not at all considered throughout this work.

Further, the conclusion of this review is just a summary of what has been stated within the single paragraphs before. At the latest here there should be some critical interpretation, future ideas, and speculations included. Is any function of ARH1 connected to its ability to reverse ADP-ribosylation of Arginines? Are there functions independent of catalytic activity? Are there any speculations about extracellularly acting arginine-specific hydrolases to counteract ARTC1-mediated ADP-ribosylation?

Line 31

ADP-ribosyl-acceptor should be ADP-ribosyl hydrolase (ARH)

Line 41

Mono- and poly-ADP-ribosylation are reversible post-translational modifications

Line 43

The amount and duration of cellular ADP-ribosylation are controlled by substrate-specific transferases and hydrolases,...

I think we are far away from understanding substrate-specificity. What dictates substrate-specificity? Besides Arginine, which is modified by ARTCs and hydrolyzed by ARH1 there are further amino acids being controversially discussed as acceptors for ADP-ribosylation. What exactly are the enzymes catalyzing one or the other and how specific are the hydrolases?

Line 66/67

One major difference between ARH1 and ARH3 is the affinity for ADP-ribose,...

Line 69

ARHs should be ARH1. ARH1 has four phosphorylation sites...

Line 76

Spacing after reference 27 is double

Line 86

... was significantly lower found in that of... should be ...was significantly lower than in that of

Line 104

Reference 20 should not be cited in this context

Line 107

Only refer to reference 20, reference 30 should not be cited in this context

Lines 121-123

This sentence is not written well und therefore not understandable

Line 134

Reference 55 should not be cited in this context. From this reference one cannot conclude the requirement of Mg2+ ions for ARH1.

Lines 135/136

ARH1, ARH3 and macrodomain proteins, ...., hydrolyze ADP-ribose from a-OAADPr.

ARH1 is a really poor hydrolase for the metabolite compared to ARH3 or the macrodomains, that are cited here.

Line 144

Spacing after increased is double

Lines 150 – 152

This sentence should be rewritten. It is not understandable

Line 167

Reference describing the double mutant (D60, 61A) is missing here. Please refer to 3. Further, one can also cite 29 in this context.

Lines 174/175

Tumor formation ... was increased in female mice and enhanced by estrogen [30, 65].

Based on ref. 30 tumor formation is increased in female mice according to percentages written in the text. However, the reference cited to demonstrate enhancement by estrogen is really poor. Unfortunately, there is absolutely no data available demonstrating this and thereby supporting this statement.

Lines 180/181

Sentence refers to ref. 30 but in this study no estrogen was used.

Line 186

In contract, ... should be in contrast, ...

Lines 207 – 209

Refer to ref. 20

Line 222

Wound healing was demonstrated using an laser... should be using a laser injury model...

Line 247

..., 33 arginine ADP-ribosylated proteins...

Looking at the according reference and counting proteins listed in the table in there, it comes to 37 arginine ADP-ribosylated proteins...

Lines 253/254

Delete the sentence: in wild-type mouse heart... have been identified.

This has already exactly been stated like this shortly before.

Then continue: Among these 354 proteins, only 6 proteins were identified...

Line 260

References are missing, where more than a thousand proteins ADP-ribosylated on arginine residues have been identified

Line 282

Just a comment: muscle weakness Ref 19, this does not necessarily be dependent on ARH1. It is stated that the identified proteins are mainly located at the cell surface or on the extracellular space, where ARH1 by localization cannot act as a hydrolase...

Reviewer 3 Report

The authors propose a review on the ADP-ribosylhydrolase ARH1, giving a particular emphasis on its cell functions.

I think this is a useful overview on the state of the art on ARH1; below just few comments to improve the quality of the text.

Pag. 1, line 37-39: please better explain the role of the aspartates;

Pag. 1, line 41: better to say ADP-ribosylation (mono- and poly-ADP-ribosylation) is a reversible…

Pag. 2, line 43: I suggest to introduce the ADP-ribosylhydrolases in wider manner (even in a concise way), also including PARG and the mono-ADP-ribosylhydrolases. After this general introduction, the focus should be more on ARH family.

Pag. 2, line 61: describe the catalytic mechanism mediated by ARH1.

Pag. 2, line 69: i) include the functional effects of phosphorylation;

ii) are similar phosphorylation sites also present in ARH3? Are kinases involved known? Did anyone perform analysis using phosphorylation site prediction tools?

iii) could be the different affinities between ARH1 and -3 for ADP-ribose explained by differences in phosphorylation?

Pag. 2, line 71, paragraph 2.2: it would be useful to describe the functional consequences of ADP-ribosylation on the targets listed.

In addition, line 85: data from Oncomine database are reported; are the same data obtained by using different cancer databases (e.g. cBioportal)?

Pag. 2, line 89-90: rephrase; the same sentence has been already used before.

Pag. 3, line 92: NAD should be NAD+; same for lines 99, 102, 104, 109 (twice).

Pag. 3, line 93: critical proteins; could you please indicate the targets?

Pag. 3, line 110: …and under ischemic stress, thus providing… (full stop after “stress” should be substituted with a comma).

The effect of ADP-ribosylation on TRIM72 should be described in more details. Try to include additional examples of proteins targets of ARH1.

Pag. 3, paragraph 2.4: the entire paragraph can be combined with paragraph 2.1.

Pag. 3, line 137: “Functions of ARH1 in diseases” and not “Function of ARH1 in diseases”

Pag. 4, line 142: substitute “alpha” with a

Pag. 4, line 148: number of proteins; list the proteins targets of the Cholera toxin.

Pag. 4, line 159: extra space before Arh1; Pag. 4, line 163: delete comma after mice;

Pag. 4, line 1637: delete comma after gene; Pag. 4, line 187: full-stop is missing after [29].

Pag. 4, line 182: an introduction on CD38 is required. Put in a context.

Pag. 5, line 219: how was the NAD+ concentration determined?

Pag. 5, line 222: …using an laser; change “an” to “a”

Pag. 6, Figure 1: a role of the stimulus activating the modification is described in the legend, while it is missing in the figure; try to include this information.

Pag. 6, line 247: ART2 has not been described as an active hydrolase in the text; explain better why now it is the enzyme used for the over-expression experiments. It sounds strange. Maybe, additional details would clarify this aspect.

Pag. 8, Conclusion paragraph: last sentence is not conclusive at all; I would add a more general sentence, with a perspective flavor.

Round 2

Reviewer 1 Report

The review has been significantly improved. There are still a number of concerns regarding the text that should be corrected. Careful proofreading will help. I specify a few:

Lane 36: “The mouse Arh1gene shows 83% identity and 91% similarity to human ARH1”. A gene does not have identity with a protein.

Lane 78: should probably read “……. and whether another ………”

Lane 109/110: ARH1 phosphorylation, …………..

Lane 145: ….., then which ………… ??

Lane 153: Ecto-ART proteins ………..

Lane 222: ………. is an NAD …………….

Lane 340: ……, probably due …………

Reviewer 2 Report

This review has improved a lot. I just have two minor comments

Line 79

Double spacing after that

Lines 197 – 202

Differ in font from the others
